# MODS: Multi-source Observations Conditional Diffusion Model for Meteorological State Downscaling

## Abstract

Accurate acquisition of high-resolution surface meteorological conditions is critical for forecasting and simulating meteorological variables. Directly applying spatial interpolation methods to derive meteorological values at specific locations from low-resolution grid fields often yields results that deviate significantly from the actual conditions. Existing downscaling methods primarily rely on the coupling relationship between geostationary satellites and ERA5 variables as a condition. However, using brightness temperature data from geostationary satellites alone fails to comprehensively capture all the changes in meteorological variables in ERA5 maps. To address this limitation, we can use a wider range of satellite data to make more full use of its inversion effects on various meteorological variables, thus producing more realistic results across different meteorological variables. To further improve the accuracy of downscaling meteorological variables at any location, we propose the **M**ulti-source **O**bservation **D**own-**S**caling Model (**MODS**). It is a conditional diffusion model that fuses data from multiple geostationary satellites GridSat, polar-orbiting satellites (AMSU-A, HIRS, and MHS), and topographic data (GEBCO), as conditions, and is pre-trained on the ERA5 reanalysis dataset. During training, latent features from diverse conditional inputs are extracted separately and fused into ERA5 maps via a multi-source cross-attention module. By exploiting the inversion relationships between reanalysis data and multi-source atmospheric variables—such as brightness temperature and precipitation—MODS generates atmospheric states that align more closely with real-world conditions. During sampling, MODS enhances downscaling consistency by incorporating low-resolution ERA5 maps and station-level meteorological data as guidance. Experimental results demonstrate that MODS achieves higher fidelity when downscaling ERA5 maps to a 6.25 km resolution, better preserving actual meteorological details.

## 1 Introduction

Global climate data assimilation provides data on the Earth's climate system at all scales, making more accurate weather forecasting possible (Lahoz & Schneider, 2014; Carrassi et al., 2018; Li et al., 2024). Among available datasets, ERA5 is the fifth-generation global climate reanalysis dataset released by the European Centre for Medium-Range Weather Forecasts (ECMWF), which offers extensive and reliable meteorological data (Hersbach et al., 2020; Bell et al., 2021). By assimilating diverse observational sources ERA5 reconstructs historical atmospheric states using advanced numerical weather prediction models and data assimilation techniques (Muñoz-Sabater et al., 2021; Jiang et al., 2021a; Yilmaz, 2023).

However, higher-resolution ERA5 data are necessary for precise weather forecasting (Cheon et al., 2024; Lopes et al., 2024). The standard ERA5 datasets, with a spatial resolution of approximately 27.75 km, lack the granularity required to accurately represent meteorological conditions in localized regions (Chattopadhyay et al., 2022). Traditional physics-based downscaling algorithms entail significant computational resource requirements. Due to this limitation, most observational data are discarded during the assimilation process, which in turn limits the diversity of observational data integrated into the assimilation process (Vandal et al., 2024; Juan et al., 2023; Giladi et al., 2021). In

recent years, numerous studies have shown that artificial intelligence (AI) models have surpassed traditional physics-based downscaling methods, achieving significant improvements in both spatial detail resolution and accuracy (Mardani et al., 2025a; Tomasi et al., 2024; Ling et al., 2024). As shown in Figure 1, current AI-based downscaling methods frequently concentrate solely on the images themselves, relying on intricate network architectures to learn the relationships between different resolutions (Behfar et al., 2024; Koldunov et al., 2024). However, directly generating high-resolution ERA5 maps through such methods often neglects the coupling relationship between ERA5 data and satellite observations, as the data in observation affects the atmospheric state in ERA5 maps, resulting in ERA5 maps that hardly conform to the real meteorological conditions of the atmosphere (Knapp & Wilkins, 2018).

To introduce satellite observations, the pioneering work of SGD (Tu et al., 2025) demonstrates that a downscaling model conditioned on satellite data—while explicitly learning the coupling relationships between these observations and ERA5 maps—can produce high-resolution (HR) ERA5 representations with improved accuracy and greater fidelity to real-world meteorological conditions. This is because ERA5 integrates a wide variety of historical observational data, particularly satellite measurements, within its advanced data assimilation and modeling systems to generate more accurate atmospheric estimates. Therefore, the atmospheric states derived from these observational sources and data from multiple origins can also be used to evaluate the generation accuracy of generative models related to ERA5 maps (Allen et al., 2025). On the one hand, apart from the GridSat data, the satellite observational datasets AMSU-A, HIRS, and MHS conduct atmospheric observations from different perspectives, such as microwave and infrared. The provided data can complement each other, enabling a more comprehensive description of the atmospheric state. Meanwhile, the atmospheric elements such as temperature and water vapor observed by AMSU-A, HIRS, and MHS are also crucial parameters influencing atmospheric physical processes. These observational data can be integrated with the atmospheric data in ERA5 to form a more accurate description of the atmospheric state (Duncan et al., 2022; Wylie et al., 2005). On the other hand, existing studies (Jiao et al., 2021; Simmons et al., 2020) have shown that the accuracy of ERA5 reanalysis products is closely associated with factors such as topography and climate classification. Therefore, more diverse and accurate satellite observational and more precise ERA5 maps complement each other.

Based on this, we propose MODS, a multi-source conditional diffusion-based downscaling model to construct variables such as brightness temperature and water vapor in the atmosphere from multi-angle data of different sources. During the training process of the diffusion model, data obtained from various sources, including geostationary satellite observation, polar orbit satellite observation, and topography data, undergo feature extraction by a pre-trained encoder, and then the features are fused using the cross-attention module. Each type of data is a different tensor with spatial, temporal, and channel dimensions. This tensor reflects a diverse range of information, including the brightness temperature of the Earth's atmosphere, water vapor content, and surface topography. Apart from static topographic data, these data provide a supplementary view of the atmosphere and enable the model to learn complex relationships across

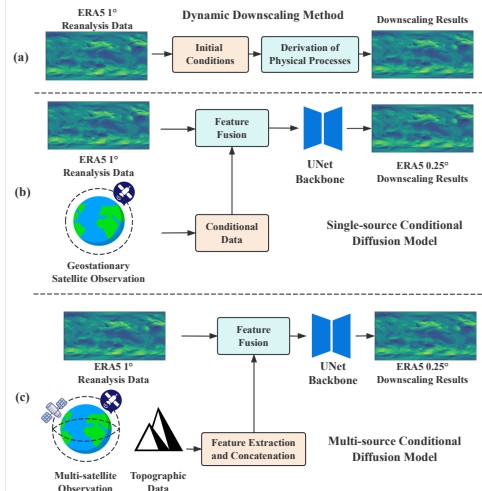

Figure 1: A paradigm shift from dynamic downscaling to multi-satellite observations conditional downscaling. Our MODS is illustrated in (c).

spaces and modalities, ensuring the authenticity of the model's generation results. In the sampling process, we use low-resolution (LR) ERA5 maps to provide guidance for the generation process. We utilize convolution to simulate the resolution conversion process, aiming to make the generated HR ERA5 maps match their corresponding LR ERA5 maps after the upscaling convolution simulation, thus guaranteeing the matching of details. Our contributions are threefold:

- In the conditional diffusion model, we have taken into account the coupling relationships existing between various satellite observational data and ERA5 maps. By using data such as brightness temperature, water vapor, and topography as conditions, the generated HR ERA5 maps are made to better conform to real-world meteorological scenarios.

- Effective feature extraction and fusion have been carried out on each meteorological data through pre-trained modules and cross-attention modules. The experimental results have verified that MODS is capable of generating higher-quality ERA5 maps.
- The sampling process of MODS can utilize diverse guidance to direct the generation process. Dynamic control of the generation can be achieved through different generation requirements, either via station-guided methods or reconstruction loss, which endows it with strong flexibility and practicality.

## 2 RELATED WORKS

**Satellite-assisted Data Assimilation.** Meteorological data assimilation is an analysis technique aiming to generate a more accurate and comprehensive representation of the Earth system, which enhances the precision of numerical simulations and forecasts in various fields like meteorology and environmental science (Valmassoi et al., 2023; Wang et al., 2024a; Cluzet et al., 2022). Nowadays, utilizing satellite observation in assimilation has been attracting increasing attention (Toth et al., 2024). Previous study (Yang et al., 2023) shows that integrating satellite radiance data into regional forecasting systems or assimilation work can provide a more comprehensive understanding of atmospheric conditions. However, traditional assimilation methods are merely confined to meteorology-related physical approaches. The emergence of diffusion models has pioneered new methods for satellite-assisted data assimilation Gong et al. (2024); Fei et al. (2023); Tu et al. (2024); Xu et al. (2024). SGD (Tu et al., 2025) is the first diffusion-based method to incorporate brightness temperature data from the GridSat satellite observation as condition in meteorological data assimilation tasks. However, considering only brightness temperature data neglects the influence of water vapor and topography data on the partial states of the atmosphere, failing to fully reflect the evolutionary characteristics of each variable in the ERA5 maps. Therefore, MODS combines geostationary satellites and multiple polar-orbiting satellites as conditions to assist in the generation of high-quality ERA5 meteorological variables.

**Meteorological Downscaling.** Downscaling aims to generate HR data from their LR counterpart, bridging resolution gaps in areas such as climate modeling and remote sensing (Kusumoto et al., 2024; Wang et al., 2024b). Traditional regression-based statistical methods leverage spatial correlations but struggle with complex nonlinear patterns (Haylock et al., 2010). Recently, deep learning methods have revolutionized this field by learning hierarchical features directly from data (Lian et al., 2024). The Generative Adversarial Networks (GANs) introduced adversarial training to generate small-scale maps that the discriminator cannot distinguish from real HR maps (Reda et al., 2023). The emergence of diffusion models, with their powerful map-generation capabilities, has provided a series of potent methods for meteorological downscaling (Mardani et al., 2025b; Merizzi et al., 2024). Latent Diffusion (Rombach et al., 2021) reduced computational costs by operating in a compressed latent space. However, when dealing with the ERA5 downscaling task, these methods overlook that the ERA5 data are derived from the inversion of multiple satellite observational sources, unable to generate high-quality HR ERA5 maps that are consistent with actual meteorological conditions.

## 3 METHODOLOGY

As shown in Figure 2, MODS aims to construct a conditional diffusion model by leveraging the coupling relationships between multi-source data such as brightness temperature, water vapor, and topography data from multi-satellite observations and ERA5 maps. This is to ensure that the HR ERA5 maps generated through downscaling are more consistent with the actual meteorological conditions. Section 3.1 elaborates the specific structure and function of the pre-trained encoder used to extract features from polar-orbiting satellite, geostationary satellite, and topography data. Section 3.2 presents and analyzes the feature fusion mechanism in the conditional diffusion model. During the sampling process, Section 3.3 elaborately derives the implementation method of MODS, which utilizes LR ERA5 maps and station-scale meteorological variables as guidance. Meanwhile, Section 3.3 analyzes the simulation effect of the convolutional kernel with optimizable parameters on the resolution conversion process and the methods of its parameter updating.

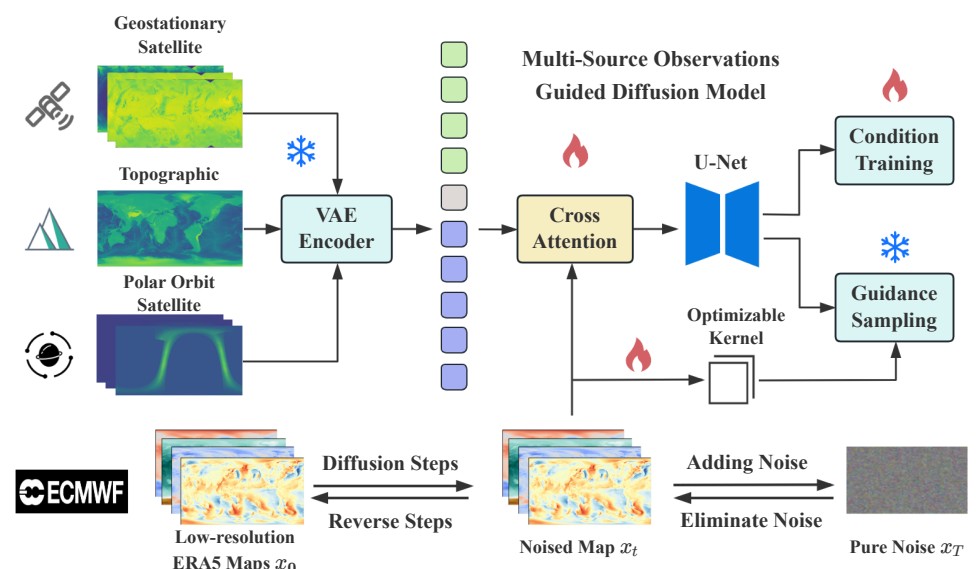

Figure 2: Overall Structure of MODS: During the training process, data from multiple satellite observations and topographic data are subjected to feature extraction via their respective VAE-based pre-trained encoders. Once these features are integrated, they serve as conditional inputs and are fused with ERA5 maps through cross-attention module, enabling the training of a conditional diffusion model. In the sampling process, LR ERA5 maps offer guidance for the reverse process. The HR ERA5 outcomes generated at each reverse step are upscaled using a convolutional kernel with optimizable parameters. Subsequently, the gradient of the distance between the upscaled results and the LR ERA5 maps is employed to update the mean for the subsequent sampling step. Simultaneously, station-scale data can be incorporated to assess the real-time accuracy of the generated HR ERA5 maps. This incorporation aids in guiding the generation process of the diffusion model from diverse perspectives, consequently improving the quality of the model's downscaling results.

## 3.1 Multi-source Observation Encoder

Before applying the cross-attention module in the MODS for conditional feature fusion, a pre-trained encoder is employed to extract the latent embedding features of various variables from satellite observations and topographic maps. The encoder extracts the features of each channel within the condition maps via three key components: the convolution layer, the down-sampling blocks, and the output blocks, while the decoder adopts a symmetric architecture to reconstruct the condition maps. The entire VAE-based structure undergoes training by leveraging the Mean Squared Error loss between the reconstructed maps and the original condition maps. For different sources of observational data, MODS pre-trained the corresponding encoder with parameter $\varphi_i$. In this way, MODS can integrate each conditional map into the latent space, thereby achieving more effective feature extraction and simplifying subsequent operations such as feature fusion. The specific structure and reconstruction results of the pre-trained modules for each condition map are presented in the Appendix.

## 3.2 Multi-conditioning Mechanisms

Diffusion models can incorporate conditional data $y$ into the diffusion process to control the generation of $x$ by training a conditional denoising autoencoder $\epsilon_\theta(x_t, t, y)$. ERA5 maps are derived from satellite observation data through methods such as assimilation. Moreover, the meteorological variables in these maps are also influenced by data like brightness temperature from satellite observations. The coupling relationship between them enables the use of satellite observation data to control the generation of high-resolution ERA5 maps. Meanwhile, in the downsampling task of ERA5 maps, if we simply train an image-to-image downscaling model as in traditional methods without adding conditional data, the low-resolution ERA5 maps lack detailed information in small-scale regions. This often leads to discrepancies between the generated data in detailed areas and the actual situation.

Therefore, we attempt to use multi-modal data, including satellite observation data from geostationary satellites (GEO), polar-orbit satellites (PO), and topographic (TOPO) data as conditions to control the MODS to generate HR ERA5 maps that are more consistent with the actual situation. A cross-attention module is incorporated to the U-Net structure to fuse the conditional information and ERA5 data (Rombach et al., 2022). Different conditional information is integrated through concatenation after passing through pre-trained encoders, which is effective for extracting multi-source conditional information $y = \{y_i\}$ from each data sources $y_i$ (Li et al.; Martin et al., 2025). This process allows the model to fully focus on each conditional data and perform weighted fusion of features according to the strength of the correlation.

Specifically, the cross-attention module achieves feature fusion through $W_Q(X), W_K(Y), W_V(Y)$ after the softmax operation, where $Y$ is obtained by concatenating the intermediate representations of multi-modal conditional data.

$$\mathbf{Y} = Concat(\varphi_a(\mathbf{y_{GEO}}), \varphi_{b_j}(\mathbf{y_{PO_j}}), \varphi_c(\mathbf{y_{TOPO}})), \tag{1}$$

$$\mathbf{Atten}(\mathbf{Q}, \mathbf{K}, \mathbf{V}) = Softmax(\frac{W_Q(\mathbf{X}) \cdot W_K(\mathbf{Y})}{\sqrt{d}} \cdot W_V(\mathbf{Y})), \tag{2}$$

where $j = 1, 2, 3$ represent the data from AMSU-A, HIRS, and MHS, respectively. The intermediate representation of each data $\epsilon_k(y) \in \mathbb{R}^{c_k \times d \times d_{\tau_k}}, k = a, b, c$ is processed through a mapping matrix with optimizable parameters. $W_Q(x)$ extracts the data from ERA5 maps, while $W_K(Y)$ and $W_V(Y)$ are obtained from the multi - modal conditional data. This structure is located at the beginning of the U-Net network. The parameters are trained and optimized along with the diffusion model, ultimately resulting in a multi-conditional diffusion model.

### 3.3 MULTI-GUIDED SAMPLING

During the sampling process, the meteorological variables from LR ERA5 maps and station observations are utilized as multi-guidance to direct the generation of HR ERA5 maps. The utilization of LR ERA5 maps aims to ensure that the results generated by the model do not differ in details from the corresponding LR maps. The incorporation of station observation measure in real-time the numerical differences of the meteorological data of the generated images at each meteorological station.

To measure the differences between the above two aspects, we introduce $\mathcal{L}_{tot} = \mathcal{L}_{res} + \mathcal{L}_{dev}$ to measure the detail difference and the numerical difference of the meteorological data at the station-scale respectively.

To avoid the problem of scale mismatch between the generation space and the input space, a scale conversion function $f_1$ is introduced to convert HR ERA5 maps into their LR counterparts. It consists of convolutions with optimizable parameters, where the stride is used to control the multiple of the scale conversion, and its parameters are also dynamically updated as the sampling progresses. Meanwhile, to calculate the meteorological data at the station-scale of the maps in the generation space, $f_2$ is introduced to calculate the meteorological data of the ERA5 maps at the point-scale. Thus, the total loss $\mathcal{L}_{total}$ during the sampling process can be expressed as:

$$\mathcal{L}_{total} = \lambda_1 \mathcal{L}_1(f_1(\tilde{x}_0), z_l) + \lambda_2 \mathcal{L}_2(f_2(f_1(\tilde{x}_0)), f_2(z_r)). \tag{3}$$

Here, $\tilde{x}_0$ represents the estimated value of the immediate output at each reverse step during the sampling process. $z = z_l, z_r$ represents the guidance data LR ERA5 maps $z_l$ and station-scale data $z_r$ respectively. $\lambda_i, i = 1, 2$ serve as weights to control the degree of incorporation of the LOSS. $\mathcal{L}_i$ acts as a distance metric, and here the Mean Squared Error (MSE) loss is used. It is worth noting that the settings of the LOSS and the distance metric here are quite flexible. Different settings can be employed according to different actual requirements, and the effect of guided sampling can also be achieved.

Previous studies (Fei et al., 2023) have derived the formula for the conditional distribution in the guided sampling process, where $N_1$ is a constant:

$$\log p_\theta(x_t|x_{t+1}, y, z) = \log (p_\theta(x_t|x_{t+1}, y)p(z|x_t)) + N_1 \tag{4}$$

The guided sampling is reflected in the introduction of the conditional distribution $p(z|x_t)$. In each reverse step, the mean and variance used are as follows:

$$\mu = \mu_\theta(x_t, y, t) + \Sigma_\theta \cdot \nabla_{x_t} \log p(z|x_t) \tag{5}$$

$$\Sigma = \Sigma_\theta(x_t, y, t) \tag{6}$$

Table 1: Data introduction of various conditional datasets.

| Dataset | Type | Acquisition | Original Shape | Intermediate Representation Shape |
|---------|------|-------------|----------------|-----------------------------------|
| HIRS | | Infrared Remote Sensing | (20,960,1920) | (160,60,120) |
| MHS | Meteorological Satellite | Microwave Detector | (5,960,1920) | (40,60,120) |
| AMSU - A | | Microwave Detector | (15,960,1920) | (120,60,120) |
| GridSat | | Satellite Observation | (3,960,2560) | (24,60,160) |
| GEBCO | Topography | Echo Probing & Satellite Altimetry | (1, 1, 1296, 2592) | (16, 60, 160) |

Here, we use a heuristic algorithm to estimate the approximate value of $\nabla_{x_t} \log P(z|x_t)$ with the gradient of the LOSS with respect to $\tilde{x}_0$:

$$\nabla_{x_t} \log p(z|x_t) \approx \nabla_{\tilde{x}_0} LOSS(\tilde{x}_0, z) \quad (7)$$

Based on this, the overall process of guided sampling is obtained. As shown in Algorithm 1, in each reverse step, the immediate output estimated value $\tilde{x}_0$ is first obtained according to the HR ERA5 maps $x_t$ in the generation space. Subsequently, substitute it into the function $f = (f_1, f_2)$ at time $t$ to calculate the deviation $LOSS_t$ at this time. The gradient of this deviation is utilized to update the mean of the sampling process, and sample $x_{t-1}$ by combining the mean and variance of the noise prediction network $\epsilon_\theta$. Meanwhile, the gradient of $Loss_{res_t}$, which represents the difference between the LR in the input space and the scale-converted HR ERA5 maps, is also used to update the parameters in $f_1$ so as to more effectively simulate the scale conversion in the subsequent reverse steps.

**Algorithm 1 Multi-guided Sampling**

**Input:** Conditional diffusion model $\epsilon_\theta(x_t, y, t)$ with conditional data $y = \{y_i\}, i = 1, 2, \ldots, n$ and its pre-trained encoder $\varphi_i$. Guidance data $z = (z_l, z_s)$. Scale conversion function $f_1$ with parameter $\vartheta$. Interpolation function $f_2$. Loss weight $\lambda = (\lambda_1, \lambda_2)$. Learning rate $l$.

**Output:** Output HR ERA5 map $x_0$.

1: Sample $x_T$ from $\mathcal{N}(0, I)$
2: $y = Concat(\varphi_1(y_1), \varphi_2(y_2), \ldots, \varphi_n(y_n))$
3: **for all** t from T to 1 **do**
4: $\quad \tilde{x}_0 = \frac{x_t}{\sqrt{\bar{\alpha}_t}} - \frac{\sqrt{1-\bar{\alpha}_t}\epsilon_\theta(x_t,y,t)}{\sqrt{\bar{\alpha}_t}}$
5: $\quad \mathcal{L}_{res_t} = \mathcal{L}_1(f_1(\tilde{x}_0), z_l)$
6: $\quad \mathcal{L}_{dev_t} = \mathcal{L}_2(f_2(f_1(\tilde{x}_0)), f_2(z_l))$
7: $\quad \mathcal{L}_{tot_t} = \lambda_1(\mathcal{L}_{res_t}) + \lambda_2(\mathcal{L}_{dev_t})$
8: $\quad \tilde{x}_0 \leftarrow \tilde{x}_0 - \frac{s(1-\bar{\alpha}_t)}{\sqrt{\bar{\alpha}_{t-1}}\beta_t}\nabla_{\tilde{x}_0}\mathcal{L}_{tot_t}$
9: $\quad \tilde{\mu}_t = \frac{\sqrt{\bar{\alpha}_{t-1}}\beta_t}{1-\bar{\alpha}_t}\tilde{x}_0 + \frac{\sqrt{\bar{\alpha}_t}(1-\bar{\alpha}_{t-1})}{1-\bar{\alpha}_t}x_t$
10: $\quad \tilde{\beta}_t = \frac{1-\bar{\alpha}_{t-1}}{1-\bar{\alpha}_t}\Sigma_\theta(x_t, y, t)$
11: $\quad$ Sample $x_{t-1}$ from $\mathcal{N}(\tilde{\mu}_t, \tilde{\beta}_t I)$
12: $\quad \vartheta_{t-1} = \vartheta_t - l\nabla_\vartheta(\mathcal{L}_{res_t})$
13: **end for**
14: **return** $x_0$

## 4 EXPERIMENT

**Data Introduction.** ERA5 (Hersbach et al., 2020) is the fifth-generation global climate reanalysis dataset released by the European Centre for Medium-Range Weather Forecasts (ECMWF). It combines model data with satellite observations and other sources of information, and generates a comprehensive set of global atmosphere, oceans, and land data with a resolution of 0.25° through techniques like data assimilation. In the experiment, the variables we used include $U_{10}$, $V_{10}$, $T_{2m}$, and $MSL$. The first two variables represent the zonal and the meridional wind speed component at a height of 10 meters, respectively. $T_{2m}$ refers to the air temperature at a height of 2 meters, and $MSL$ stands for mean sea-level pressure. **Conditional Data.** Table 1 presents the types and sizes of data integrated as conditional data into the condition diffusion model. Among them, AMSU-A, HIRS, and MHS are carried by the NOAA-18 and NOAA-19 satellites, which operate in low-Earth orbits. GridSat data is sourced from the NOAA satellite series and is obtained through the processing of satellite data. Each of the datasets is measured by sensors carried on satellites, which provide distribution information such as atmospheric humidity and temperature, enabling MODS to gain a more comprehensive understanding of the physical state of the atmosphere. GEBCO data offers global-scale topographic data of land and ocean. More detailed data descriptions for conditional data are placed in the Appendix.

### 4.1 IMPLEMENTATION DETAILS.

For the pre-trained encoder, we pre-train it using AMSU-A, MHS, GridSat, and HIRS data with a 6-hour interval from 2010 to 2023, with a time error within 30 minutes. During pre-training, the Mean Squared Error between the reconstruction results and the original observed maps is utilized as the training loss, and the number of epochs used is 10. Regarding the topographic data GEBCO, since the differences in topographic data across different time periods are not significant, we use

Table 2: Results of MODS and off-the-shelf downscaling methods on $U_{10}$, $V_{10}$, $T_{2M}$ and $MSL$. Different colors represent different types of methods. A mixture of the LR ERA5 maps and station-scale data is utilized as multi-guidance in MODS.

| Methods | $U_{10}$ | | $V_{10}$ | | $T_{2m}$ | | $MSL$ | |
| --- | --- | --- | --- | --- | --- | --- | --- | --- |
| | MSE | MAE | MSE | MAE | MSE | MAE | MSE | MAE |
| ERA5 1° | 53.18 | 5.95 | 38.51 | 4.95 | 216.27 | 11.39 | 470.06 | 15.78 |
| Bilinear | 56.55 | 6.16 | 37.94 | 5.03 | 198.40 | 11.26 | 464.51 | 16.05 |
| Bicubic | 55.74 | 6.11 | 37.88 | 4.98 | 201.03 | 11.15 | 458.82 | 15.97 |
| IDW | 56.73 | 6.12 | 37.34 | 4.97 | 193.71 | 10.66 | 438.32 | 15.76 |
| Kriging | 55.04 | 5.87 | 37.21 | 4.95 | 187.32 | 10.51 | 420.57 | 15.48 |
| R-Kriging | 53.78 | 5.63 | **37.05** | **4.90** | 183.59 | 10.32 | 408.55 | 15.02 |
| SwinRDM (Chen et al., 2023) | 53.18 | 5.95 | 38.51 | 4.95 | 216.27 | 11.39 | 470.06 | 15.78 |
| Ref-SR (Huang et al., 2022) | 62.72 | 6.15 | 43.12 | 5.17 | 195.42 | 11.02 | 395.42 | 15.10 |
| $C^2$-Matching (Jiang et al., 2021b) | 65.12 | 6.02 | 44.57 | 5.41 | 200.17 | 11.36 | 410.72 | 15.32 |
| DGP (Pan et al., 2021) | 97.02 | 7.96 | 47.52 | 5.07 | 214.58 | 11.94 | 529.72 | 18.54 |
| GDP (Fei et al., 2023) | 94.99 | 7.85 | 40.17 | 5.04 | 190.11 | 10.82 | 511.71 | 18.14 |
| DDNM (Wang et al., 2022) | 52.08 | 5.94 | 42.17 | 5.65 | 193.24 | 11.77 | 396.58 | 15.12 |
| HyperDS (Liu et al., 2024) | 53.72 | 6.01 | 41.37 | 5.26 | 191.83 | 10.87 | 384.72 | 14.72 |
| SGD (Tu et al., 2025) | 52.03 | 5.93 | 39.82 | 5.05 | 187.69 | 10.63 | 374.39 | 14.49 |
| MODS | **43.43** | **5.36** | 39.96 | 5.05 | **155.64** | **9.42** | **371.28** | **14.20** |

single-frame data for pre-training. For the training process of MODS, the hourly ERA5 maps from 2010 to 2021 with a shape of (720×1440) are used as the training set, the data from 2021 to 2022 as the validation set, and the data from 2023 as the test set. All the training tasks are conducted on an NVIDIA A100 80GB GPU. The entire training process consists of 200,000 steps and lasts for 2-3 days. MODS is optimized using AdamW with $\beta_1 = 0.9$ and $\beta_2 = 0.999$ in 16-bit precision with loss scaling, while maintaining 32-bit weights, Exponential Moving Average (EMA), and optimizer state. Each layer of the U-Net contains 2 residual blocks, and the UNet structure has a total of 4 layers. Meanwhile, the diffusion variance $\beta_t$ is set as hyperparameter increasing linearly from $\beta_1 = 10^{-4}$ to $\beta_T = 0.02$. During the sampling process, the kernel of the scaling transformation function is $9 \times 9$, which is implemented utilizing group convolution.

## 4.2 EVALUATION METRICS.

We utilized station-scale data to examine the differences between the values of all four variables in the downscaled ERA5 maps and the actual meteorological values at various stations on a global scale. The actual meteorological variable data at the stations were sourced from the Weather 5k dataset (Han et al., 2024). This dataset is a large-scale time-series dataset for sparse weather forecasting, which encompasses 5,672 weather stations distributed across the globe. The comparison metrics included the Mean Absolute Error (MAE) and the Mean Squared Error (MSE), which are employed to quantify the discrepancies between the output results and the actual values. We extracted the values of variables at the weather 5k stations from the small-scale ERA5 maps of different methods by using the $grid\_sample$ function in PyTorch.

## 4.3 MAIN RESULTS

To measure the difference between the results of the downscaling task of MODS on ERA5 maps and real-world meteorological scenarios, we compare the metrics and results with off-the-shelf methods. Existing methods include interpolation-based methods, diffusion-based methods, and other methods that can be utilized to solve super-resolution or down-scaling tasks. As shown in Table 2, MODS outperforms existing methods in all metrics of $U_{10}$, $T_{2M}$, and $MSL$, and also shows certain improvements compared with direct interpolation methods. Notably, compared with SGD, which only uses single-source GridSat satellite observation data as a condition, MODS has advantages in all metrics. In particular, the improvement in the $T_{2M}$ metric, which reflects temperature data, is particularly significant. This verifies that when multi-source observation data is used as a condition, brightness temperature data from different sources can effectively enhance the model's ability to accurately generate the meteorological variable of temperature.

Figure 3 visually presents a comparison of images for various variables in the downscaled ERA5 maps at multiple time stamps. Compared with off-the-shelf methods, MODS exhibits a lower degree

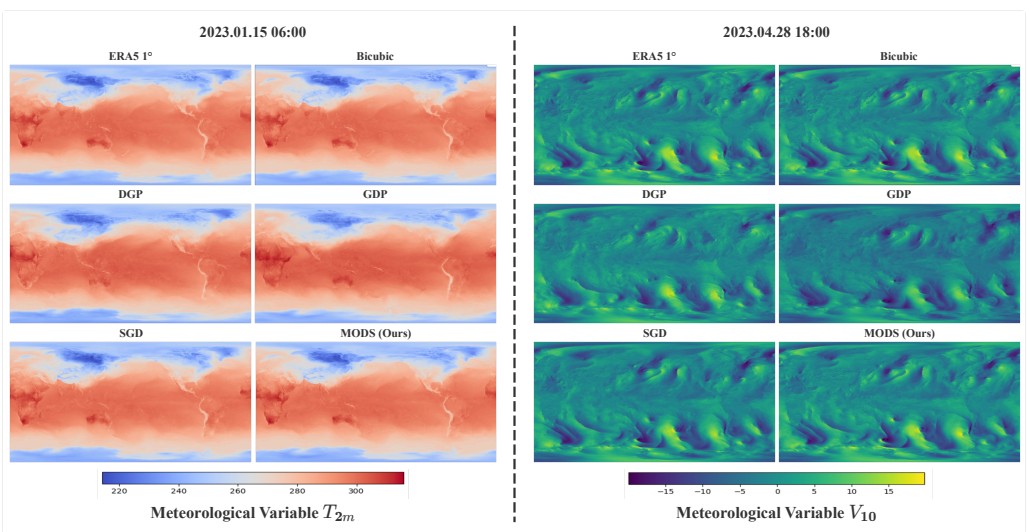

Figure 3: Comparison of the downscaling results of ERA5 maps obtained by MODS with other methods such as interpolation-based and diffusion-based methods.

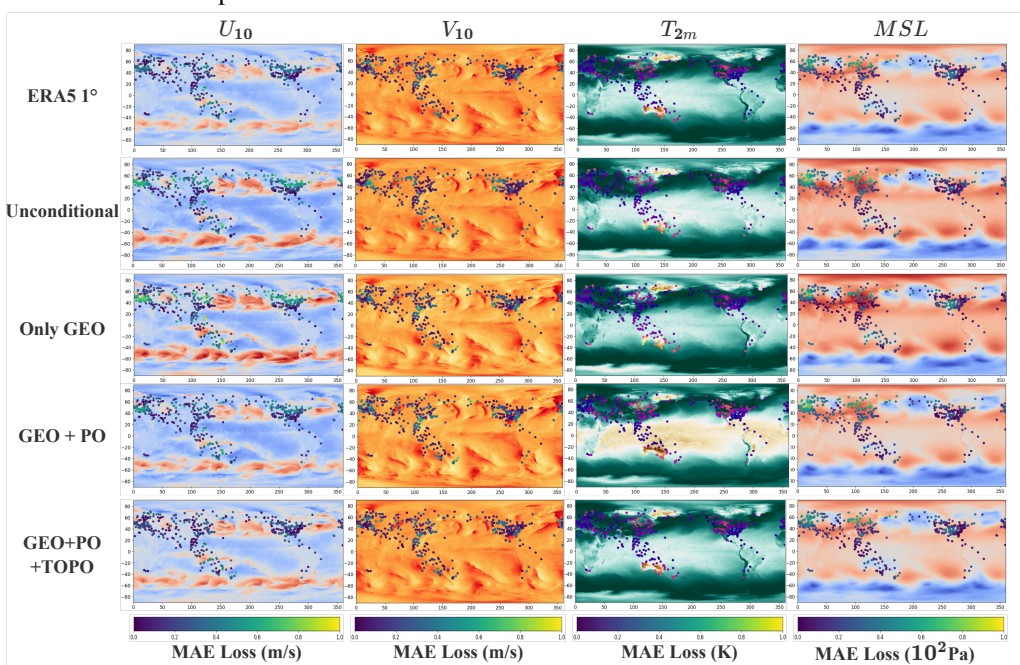

Figure 4: Comparison of meteorological differences at the station-scale when using different data as conditions. The color of each point represents the value difference between the results and the real meteorological data at the Weather 5K stations. Brighter colors indicate larger value differences.

of detail loss. This is attributed to the use of LR ERA5 maps as guidance to dynamically control the details in each step of the reverse process. More comparison results on other variables and other time stamps are shown in the Appendix.

## 4.4 ABLATION STUDY

**The effectiveness of MODS in different condition datasets.** To evaluate the role of the multi-source data used by MODS, including the topographic data (TOPO), Geostationary Satellite Observation data (GEO), and Polar-orbiting satellite data (PO), we conduct an ablation study to examine the metric results when only part of this data is utilized. Water vapor and brightness temperature data from satellite observations are factors influencing the atmospheric state. Multi-source data can

Table 3: Ablation study on the downscaling results of MODS when using different satellite data as conditions.

| Methods | Conditions | $U_{10}$ | | $V_{10}$ | | $T_{2M}$ | | $MSL$ | |
|---|---|---|---|---|---|---|---|---|---|
| | | MSE | MAE | MSE | MAE | MSE | MAE | MSE | MAE |
| Partial-conditional or unconditional MODS | Uncondition | 74.02 | 6.87 | 47.11 | 5.52 | 208.72 | 11.15 | 564.50 | 18.67 |
| | GEO | 51.42 | 5.76 | 40.18 | 5.12 | 186.05 | 10.60 | 373.85 | 14.51 |
| | PO | 53.04 | 5.81 | 40.01 | 5.10 | 189.51 | 11.03 | 390.56 | 15.02 |
| | TOPO | 70.58 | 6.44 | 46.36 | 5.17 | 202.60 | 11.05 | 507.52 | 18.11 |
| | GEO & PO | 46.72 | 5.61 | 39.98 | **5.03** | 157.92 | 9.61 | 371.96 | 14.33 |
| | GEO & TOPO | 49.83 | 5.81 | 40.06 | 5.07 | 176.82 | 10.51 | 372.09 | 14.39 |
| | PO & TOPO | 51.94 | 5.78 | 39.96 | 5.05 | 181.70 | 10.81 | 381.54 | 14.76 |
| Multi-conditional MODS | GEO & PO & TOPO | **43.43** | **5.36** | **39.96** | 5.05 | **155.64** | **9.42** | **371.28** | **14.20** |

Table 4: Ablation study on the different types of data as guidance in MODS sampling process.

| Methods | $U_{10}$ | | $V_{10}$ | | $T_{2M}$ | | $MSL$ | |
|---|---|---|---|---|---|---|---|---|
| | MSE | MAE | MSE | MAE | MSE | MAE | MSE | MAE |
| MODS guided by LR ERA5 | 60.84 | 7.01 | 45.86 | 6.40 | 201.87 | 11.21 | 408.85 | 15.32 |
| MODS guided by station-scale data | **43.43** | **5.36** | **39.96** | **5.05** | **155.64** | **9.42** | **371.28** | **14.20** |
| Multi-guided MODS | 54.32 | 5.82 | 40.52 | 5.66 | 172.04 | 10.12 | 374.26 | 14.49 |

simulate and reconstruct the real-world atmospheric state from multiple dimensions. The results in Table 3 validate the effectiveness of this setting. Compared to using only part of the multi-source data as conditions, MODS achieves ERA5 downscaling results that are closer to real-world conditions for all meteorological variables. As shown in Figure 4, the comparison also reveals that the brightness temperature data from GEO has a relatively greater impact on the results, especially on the corresponding $T_{2M}$ metric. In contrast, the TOPO data has a relatively smaller but more comprehensive impact on the results.

**The effectiveness of MODS utilizing different guidance during sampling.** During the sampling process of MODS, the weights of different parts in the loss function can be flexibly adjusted to modify the types of data added as guidance. By incorporating LR ERA5 maps as part of the guidance, MODS can prevent the generation of HR ERA5 corresponding maps that suffer the loss of details compared to the LR ERA5 maps. Utilizing station-scale meteorological station data enables real-time measurement and control of the differences between the generated meteorological data at various stations and the actual conditions. By assigning different weights to the above two data sources, multi-guided sampling can be achieved. As shown in Table 4, using station observations and LR ERA5 maps as multi-guidance helps MODS generate HR ERA5 maps that are more consistent with actual meteorological scenarios. This effect is more prominent when station observations are solely used as guidance. However, an excessive focus on the meteorological data at stations may also lead to the loss of details in the generated results.

## 5 CONCLUSION

The proposed model MODS is a downscaling model for ERA5 data based on a multi-source conditional diffusion model. It integrates three different types of polar-orbiting satellite data, including AMSU-A, HIRS, and MHS, along with geostationary satellite data GridSat and topographic data GEBCO as multi-source conditions. The brightness temperature, water vapor, and topographic data in the multi-source data are key parameters influencing the physical state of the atmosphere. Moreover, the data from different sources can complement each other, enabling a more comprehensive description of the atmospheric state. Meanwhile, the ERA5 data also incorporates satellite observation data in its assimilation system to achieve a more accurate estimation of the atmospheric state. Based on this, during the training process, the multi-source data undergoes feature extraction through a pre-trained encoder structure. Then, a cross-attention structure is employed to fuse the features of the multi-source data with the ERA5 maps. This enables MODS to generate ERA5 maps that are more in line with real-world meteorological condition during the sampling process. In the sampling process, LR ERA5 maps and station observations are utilized as multi-guidance to dynamically control the loss of details and the differences in meteorological states in the generated results. Experiments have verified that the setting of using multi-source data as conditions in MODS can generate high-quality HR ERA5 maps with faithful details and more realistic meteorological variables.

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
