## A    COMPARISON OF RECONSTRUCTION RESULTS

Before the multi-source conditional data is integrated into the MODS model, it is necessary to perform corresponding pre-training on each conditional data to extract its intermediate representations. The overall structure of the pre-trained encoder and decoder is shown in Figure 5. MODS adopts an Encoder-Decoder structure based on VAE. It utilizes five down-sampling blocks and up-sampling blocks, respectively, to extract features from each dataset and reconstruct the data. To verify the effectiveness of the pre-trained model, we compared the reconstruction results of the model with the original observed maps. As shown in Figure 6, there are no discernible differences in details between the reconstruction results of the pre-trained models for each satellite observation and topographic data and their respective inputs. Meanwhile, there is no significant degradation in quality, such as blurring. This indicates that the pre-training is relatively sufficient, and the existing model can achieve effective feature extraction and reconstruction.

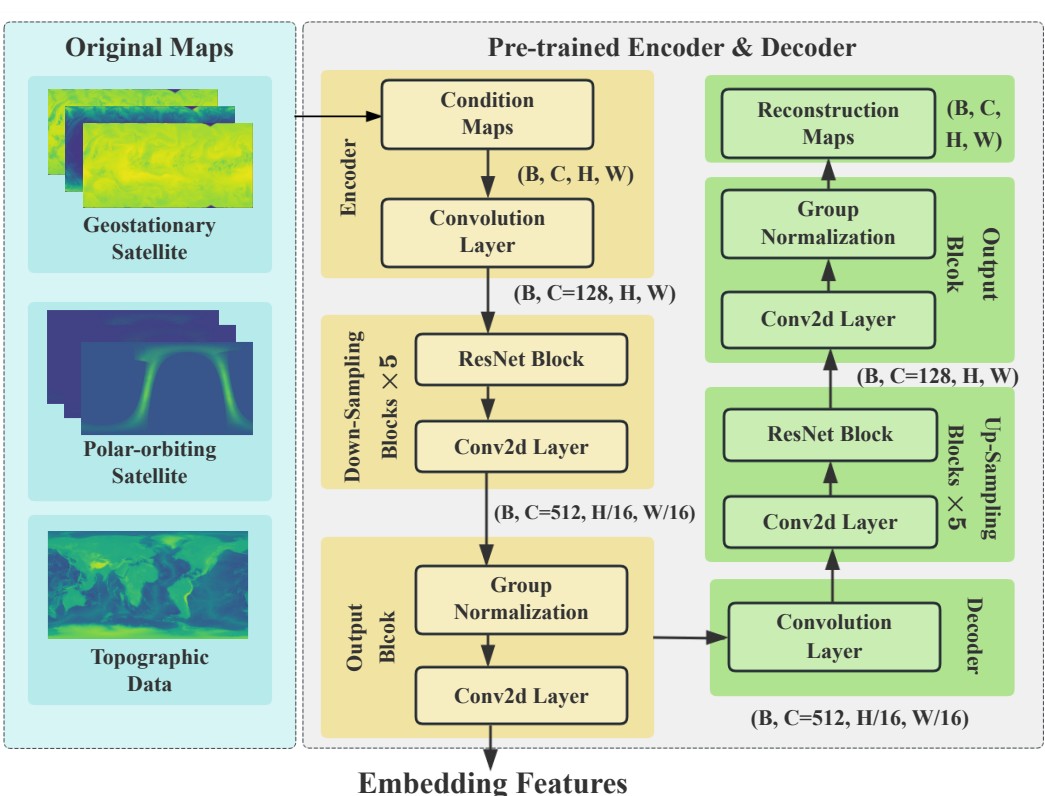

Figure 5: Overall structure of the pre-trained encoder and decoder module utilized in MODS.

## B    CONDITION DATA

**GridSat** dataset (Skofronick-Jackson et al., 2015) is derived from the observational data in visible, infrared and other bands collected by multiple meteorological satellites from the NOAA series. These data are then comprehensively processed and gridded. The data content records meteorological parameters such as cloud-top temperature and radiance in a regular grid format. The cloud-top temperature reflects the thermal state, while the radiance reflects the radiation characteristics of the Earth's surface and the atmosphere.

**HIRS** (High-Resolution Infrared Radiation Sounder) (Shi & Bates, 2011) is an instrument installed on meteorological satellites, which is used to measure the radiation of the Earth's atmosphere in the infrared band. The data obtained by these instruments are processed to form the HIRS dataset. The instrument data mainly provides information such as the vertical temperature profile of the

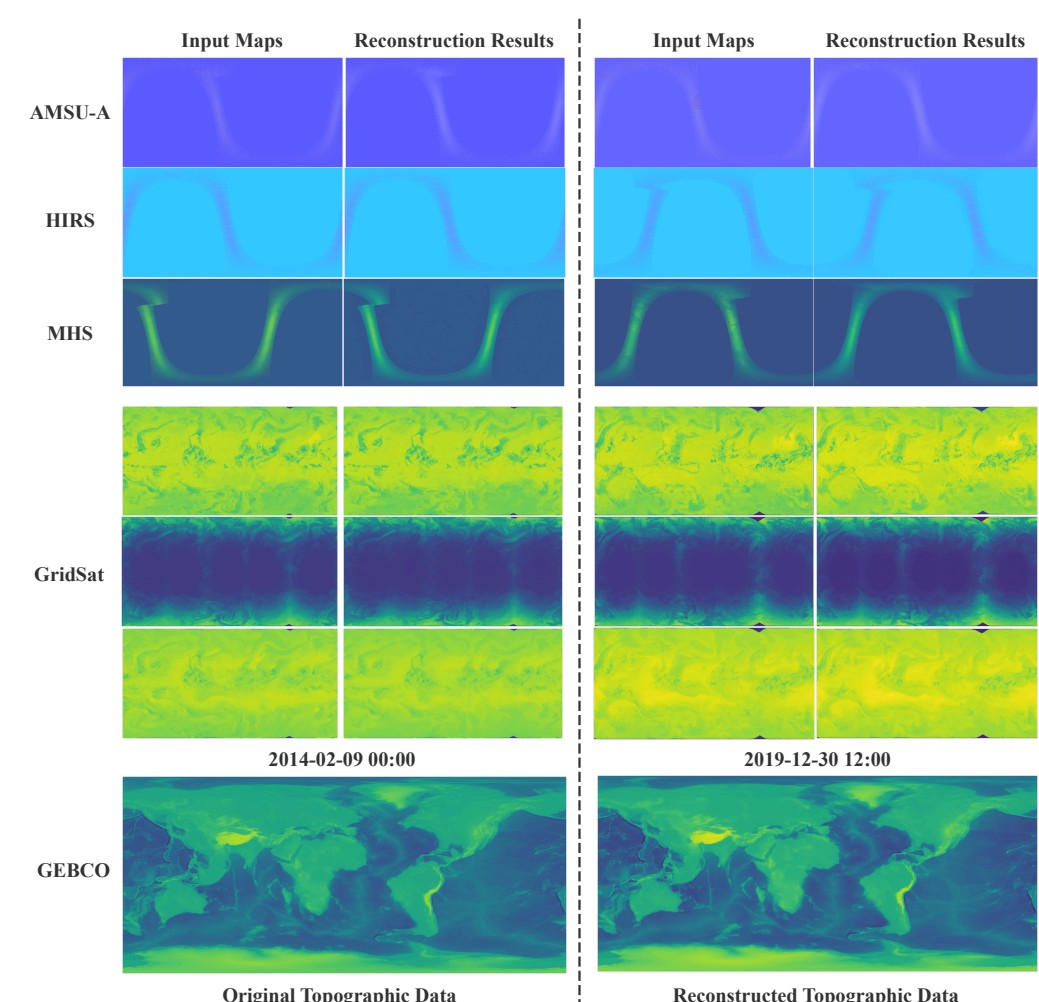

Figure 6: Comparison of input and MODS reconstruction maps for different data channels.

atmosphere. By measuring infrared radiation of different wavelengths and using the principle of radiative transfer, the temperature and water vapor conditions at different altitudes in the atmosphere can be retrieved.

**MHS** (Microwave Humidity Sounder) (Bonsignori, 2007) is installed on meteorological satellites and is used to measure the water vapor distribution in the atmosphere. The MHS instrument utilizes the absorption characteristics of water vapor in the microwave band and measures microwave radiation to retrieve the water vapor content in the atmosphere.

**AMSU-A** (Advanced Microwave Sounding Unit-A) (Mo, 1996) is a microwave detection instrument installed on meteorological satellites, which is mainly used to measure the temperature distribution of the atmosphere. By measuring microwave radiation at different frequencies, it can obtain temperature information at different altitudes in the atmosphere.

**GEBCO** (General Bathymetric Chart of the Oceans) (Mayer et al., 2018) terrain dataset is sourced from measurements by ship-borne echo sounders, satellite altimetry data, etc. The data includes global DEM data from grid-scale to watershed-scale. It combines grid DEM with high-resolution satellite remote-sensing images and divides the global land and ocean areas.

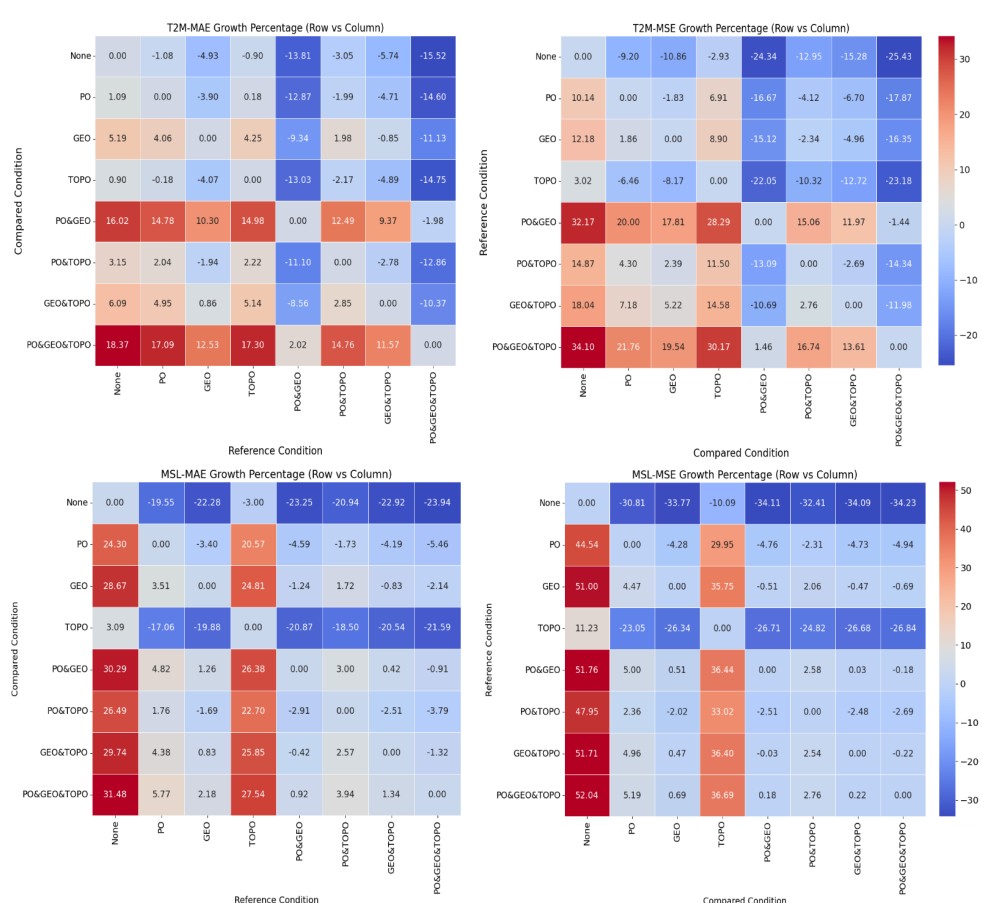

Figure 7: Heatmap figure of the relative error losses of different conditional data for variables $T_{2M}$ and $MSL$. The horizontal and vertical axes represent the types and quantities of introduced conditions. The values in the heatmap represent the percentage improvement of the MSE and MAE results of the model under the horizontal-axis conditions for this variable compared with those under the vertical-axis conditions.

## C  COMPARISON REGARDING THE INFLUENCE OF DIFFERENT CONDITIONAL DATA

To validate the impact of different data in multi-conditional data on the downscaling results of ERA5 maps, the main text conducts a quantitative comparison. The results show that the introduction of multi-conditional data has a relatively large impact on the variable $T_{2M}$, followed by $MSL$. The heatmap in Figure 7 illustrates the influence of topographic data (TOPO), Geostationary Satellite Observation data (GEO), and Polar-orbiting satellite data (PO) on the results of these two variables. Through comparison, it can be found that GEO data has the most significant improvement effect on the results compared with other data. This is because the brightness temperature data in GridSat is a crucial factor affecting the atmospheric state. Moreover, PO also has a relatively obvious impact on the results, suggesting that variables such as water vapor content influence both temperature variable $T_{2M}$ and the pressure variable $MSL$. Interestingly, combining GEO and PO yields results comparable to those of MODS, highlighting their synergistic effect. In contrast, topographic data (represented by GEBCO) has a comparatively minor influence, though it still contributes measurably to the downscaling results.

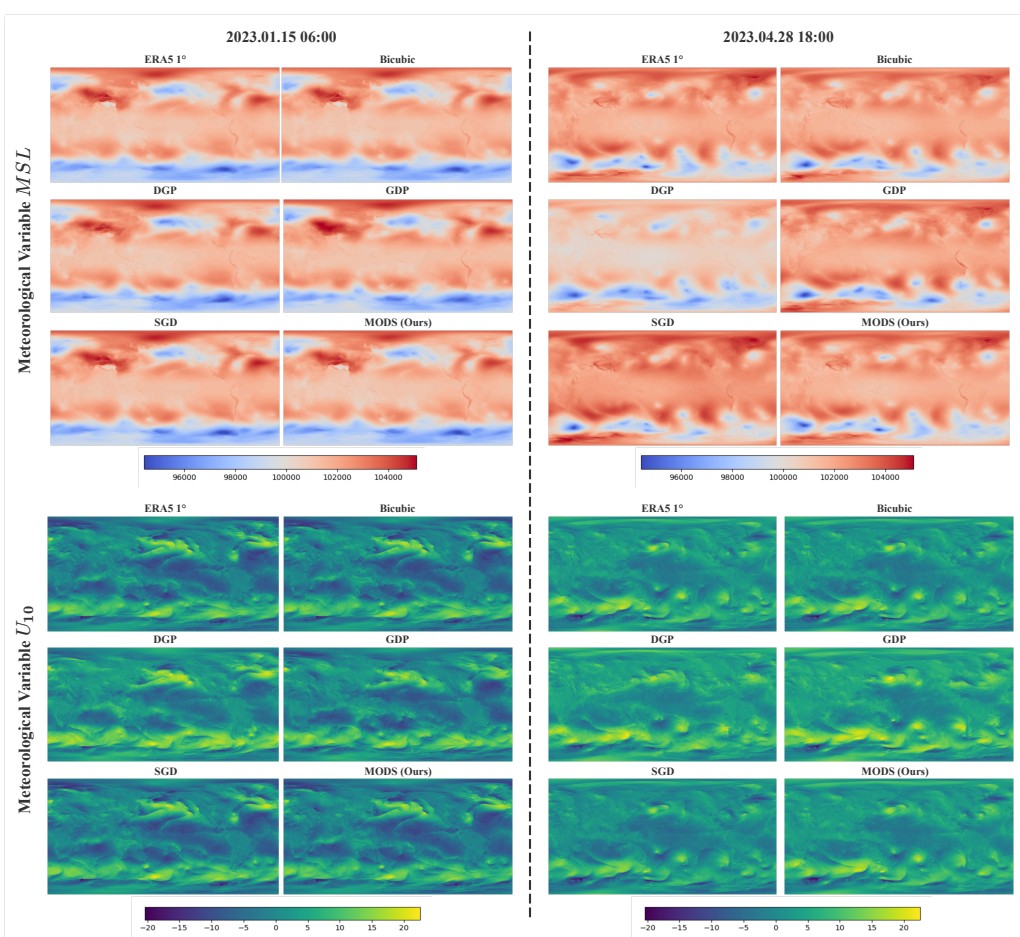

Figure 8: Extra comparison of the map results between MODS and off-the-shelf methods in the ERA5 downscaling task.

## D  LIMITATION AND FUTURE WORK

The conditional data utilized in the MODS includes topographic data, geostationary satellite observation data, and polar-orbiting satellite observation data. However, direct observations of wind speed, an important variable, are absent from these satellite observations. For instance, incorporating satellite data from ASCAT, which is related to wind speed, may further enhance the performance of MODS in estimating wind-speed-related variables such as $U_{10}$ and $V_{10}$.

## E  EXTRA COMPARISON

This section presents an extra comparison of the results between MODS and other methods across four variables. As shown in Figure 8, compared with interpolation-based and other diffusion-based methods, the small-scale ERA5 maps obtained by MODS not only feature rich and faithful details but also exhibit no obvious differences in details when compared with their LR counterparts. This indicates that MODS can generate HR ERA5 maps that are more consistent with real-world meteorological scenarios through multi-source conditions and a multi-guidance sampling strategy.

Table 5: The comparison of running time and resource consumption.

| Methods | GPU Memory | Time |
|---------|------------|------|
| GDP | $\approx 2.6 \times 10^4$ MB | $\approx 6$ min |
| DDRM | $\approx 4.3 \times 10^4$ MB | $\approx 1$ min |
| DDNM | $\approx 3.7 \times 10^4$ MB | $\approx 1$ min |
| SGD | $\approx 1.9 \times 10^4$ MB | $\approx 5$ min |
| MODS | $\approx 1.9 \times 10^4$ MB | $\approx 5$ min |

## F  RUNNING TIME AND COMPUTING RESOURCES

All training and sampling tasks of MODS were carried out on an NVIDIA A100 80GB GPU. During the training process, the entire training lasted approximately three days for about 200000 steps on the training set. For the conditional data, the training process directly took the intermediate representations obtained from its corresponding pre-trained encoder module as input. To avoid excessive computational resource consumption that would result from directly training a conditional diffusion model in HR space, MODS employs patch-based methods, which adopts a patch segmentation approach to reduce the required model scale, enabling training to be conducted on LR ERA5 maps at a scale of 25km. The overall computing resource consumption was about $4.8 \times 10^4$ MiB. During the sampling phase, patch-based methods combined with convolutional kernels with optimizable parameters are utilized to achieve effective resolution conversion. For downscaling tasks that require precision, the sampling time of MODS is acceptable for operational use. As shown in Table 5, in the sampling process, it took approximately 7 minutes to perform the downscaling task for each ERA5 map. The overall computing resource consumption was about $1.9 \times 10^4$ MiB.

## G  SPECTRAL ANALYSIS

We test the differences in the zonal power spectra between the output results of MODS and the ground truth (GT). As shown in Figure 9, For all variables, the power spectra of the model outputs and GT almost overlap in the low-wavenumber interval without typical spectral gap. In the high-wavenumber interval, the results of MODS and GT on $U_{10}$, $T_{2m}$ and $MSL$ are close. From the low wavenumbers to the high wavenumbers, the powers of both the model and GT show an approximately logarithmic decay as the wavenumber increases, which is also in line with the overall law of atmospheric waves. Since the image cannot be placed here, we compare the wavenumber of each variable in the output results of MODS and GT. As shown in the table, the deviations of each variable within different wavenumber ranges are relatively small.

## H  MORE RESULTS ON WEATHERBENCH

We further evaluated the model's performance on the Weatherbench dataset. Specifically, we selected the 2023 subset of WeatherBench at 1.5° resolution and adopted the same evaluation metric methods as those used for the Weather5k dataset. As shown in Table 6, to thoroughly evaluate the performance of MODS, we conducted tests using the weather 5k station-scale data and low-resolution ERA5 maps as guidance. MODS outperformed existing SR-based and diffusion-based methods across most evaluation metrics (the top 3 methods were selected).

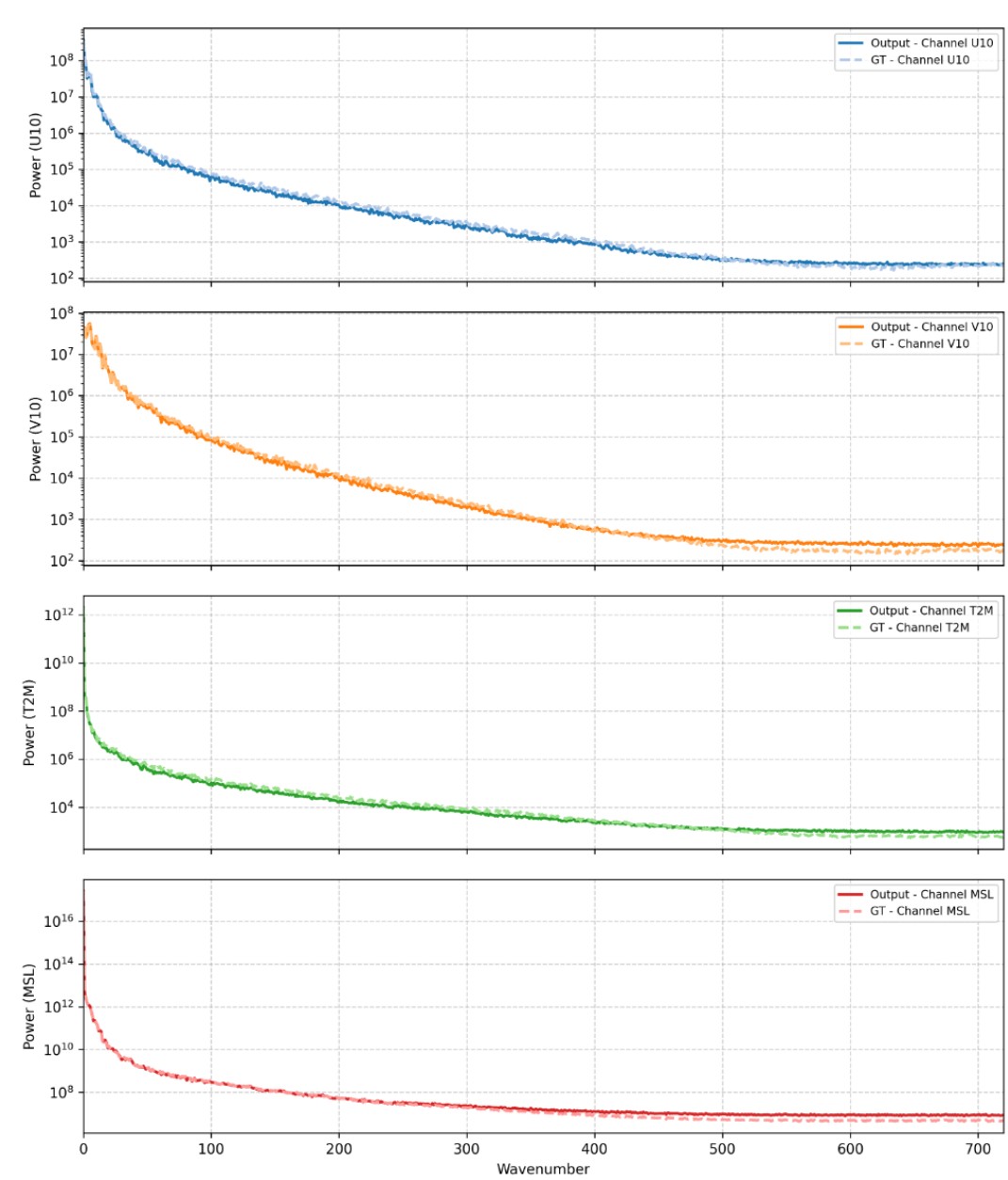

Figure 9: Zonal power spectrum comparison between MODS output and the ground truth.

Table 6: Results of MODS and off-the-shelf downscaling methods on $U_{10}$, $V_{10}$, $T_{2M}$ and $MSL$ on WeatherBench.

| Methods | $U_{10}$ | | $V_{10}$ | | $T_{2m}$ | | $MSL$ | |
|---|---|---|---|---|---|---|---|---|
| | MSE | MAE | MSE | MAE | MSE | MAE | MSE | MAE |
| SwinRDM | 22.65 | 4.96 | 19.82 | **4.27** | 130.82 | 9.75 | 380.51 | 14.47 |
| Ref-SR | 23.75 | 5.70 | 23.52 | 5.01 | 137.55 | 10.91 | 393.52 | 14.76 |
| $C-2$-Matching | 24.03 | 5.96 | 23.76 | 5.12 | 142.75 | 11.08 | 402.71 | 14.88 |
| HyperDS | 22.47 | 4.92 | 20.08 | 4.65 | 131.11 | 10.14 | 383.19 | 14.39 |
| SGD | 22.53 | 5.02 | 19.84 | 4.47 | 125.37 | 9.56 | 377.50 | 14.07 |
| MODS | **21.57** | **4.83** | **19.72** | 4.35 | **114.72** | **9.03** | **364.72** | **13.85** |