# OpenReview forum: "MODS: Multi-Source Observation Conditional Diffusion Model for Meteorological State Downscaling"
_ICLR.cc/2026/Conference — Submitted to ICLR 2026_

### Official Review · Reviewer_YwDg · 2025-10-24

**Soundness:** 3
**Presentation:** 3
**Contribution:** 2
**Rating:** 2
**Confidence:** 4

**Summary:**

The paper introduces the Multi-source Observations Conditional Diffusion Model (MODS), a framework designed to enhance meteorological state downscaling. It integrates data from multiple satellite sources and topographic information to generate high resolution ERA5 maps that better represent real world atmospheric conditions. MODS employs a conditional diffusion model that fuses extracted features through a cross attention mechanism, allowing the system to learn complex relationships among brightness temperature, water vapor, and terrain data. During sampling, the model uses low resolution ERA5 maps and station observations as guidance to ensure realistic and detail preserving outputs

**Strengths:**

- It was designed to utilize a wider range of weather variables/modalities and sensor data compared to previous studies, allowing complementary use of information that a single modality alone cannot provide.

- MODS employs a flexible guided sampling strategy using both low-resolution ERA5 maps and station data. This dual guidance ensures consistent high-resolution outputs that maintain physical plausibility across varying meteorological conditions

**Weaknesses:**

- The study shares a very similar overall task and concept with SGD [1], which it uses as a reference. Although this paper incorporates a wider range of modalities and achieves improved performance compared to the referenced work, it does not present particularly novel technical contributions.

- The study lacks domain-specific modifications as a Climate AI research work. The encoder, conditioning, and sampling methods presented are modules and approaches commonly used in the general machine learning field, typically assuming an RGB domain. These methods are likely to overlook important characteristics unique to non-RGB data such as ERA5 and various satellite sensor inputs, yet the study does not address such sensor- or domain-specific considerations. Overall, the work appears to focus mainly on extending existing machine learning methodologies and adding modalities, making it difficult to identify clear novelty in either problem definition or technical contribution.

[1] Tu, Siwei, et al. "Satellite observations guided diffusion model for accurate meteorological states at arbitrary resolution." Proceedings of the Computer Vision and Pattern Recognition Conference. 2025.

**Questions:**

- Please explain in detail under which specific scenarios each data source produces accurate or inaccurate predictions. Although Table 3 and Figure 4 present ablation studies for each data source, the current section “The effectiveness of MODS in different condition datasets” does not clearly describe which strengths of each data source contribute to robustness in particular scenarios. A more detailed explanation is needed.

- Why are the SGD results shown in Table 2 of the SGD paper different from the SGD results presented in Table 2 of your manuscript? Moreover, MODS shows relatively weaker performance in the V10 variable compared to other models. What are the technical reasons for this (for example, aspects of the framework structure or sensor characteristics)?

---

### Official Review · Reviewer_3PAF · 2025-10-31

**Soundness:** 3
**Presentation:** 3
**Contribution:** 3
**Rating:** 6
**Confidence:** 2

**Summary:**

The paper introduces MODS, a conditional diffusion model designed to generate high-resolution (6.25 km) meteorological maps from low-resolution (27.75 km) ERA5 data. Its primary innovation is twofold: 1) during training, it fuses data from multiple observation sources—including geostationary, polar-orbiting, and topographic data—using a cross-attention module to create a comprehensive conditional input ; and 2) during sampling, it uses a dual-guidance mechanism from both the original low-resolution ERA5 map and real-world station-level data to ensure the final output is both consistent and accurate.

**Strengths:**

1. Proposed method fuses complementary data from geostationary, polar-orbiting, and topographic sources, which is a clear improvement over single-source conditional methods.
2. The multi-guidance sampling strategy is robust, using both the original low-resolution map for large-scale consistency and real-world station data to constrain the model for point-specific accuracy.
3. The paper is supported by strong experimental validation, including two thorough ablation studies that systematically prove the value of both the multi-source conditions and the multi-guidance sampling.

**Weaknesses:**

1. Authors should provide analysis of MODS inference speed or cost, which is a critical omission for a complex, iterative diffusion model.
2. What is $f_2$ exactly?

**Questions:**

1. Does the baseline methods (especially diffusion-based and other learning based methods) use the same conditional data? If multiple conditions are only used in MODS, is it a fair comparison?
2. Is Scale conversion function $f_1$ randomly initialized during sampling? In classifier-guided image generation, people use a pretrained image classifier. However, in your sampling procedure, there are learnable parameters (and you are optimizing it). Can it introduce noise if it is randomly initialized? How does the learning rate l affect sampling quality?

---

### Official Review · Reviewer_nPtT · 2025-11-01

**Soundness:** 2
**Presentation:** 2
**Contribution:** 2
**Rating:** 4
**Confidence:** 5

**Summary:**

The paper proposes MODS, a conditional diffusion model for downscaling ERA5. MODS pre-trains VAE encoders for multiple exogenous inputs—geostationary & polar-orbiting satellites, and topography—and fuses them with ERA5 features via multi-source cross-attention in a U-Net backbone. At sampling time, MODS introduces multi-guided inference: a learnable downscaling kernel enforces consistency with LR ERA5, while station observations provide point-wise guidance; both are incorporated by adding a gradient term to the reverse diffusion mean. Experiments on ERA5 report improved MAE/MSE across U10, V10, T2m, and MSL versus interpolation, reference SR, and prior diffusion baselines; ablations suggest multi-source conditioning and station/LR guidance each contribute.

**Strengths:**

- Problem importance & scope. High-resolution meteorological fields are crucial for downstream modeling; integrating satellite modalities and topography is well-motivated and practically useful.

- Technical clarity (core idea). The multi-source conditioning via VAE encoders + cross-attention is a straightforward, modular way to fuse heterogeneous sensors while keeping the denoiser architecture standard.

- Ablations. Sensible “turning off” studies (GEO/PO/TOPO subsets; guidance variants) support the claim that each ingredient helps.

**Weaknesses:**

**Novelty over closest work.**

* **SGD (Tu et al., 2025)**: Very close in framing—conditional diffusion trained on ERA5 with **GridSat** as condition, plus **learnable up/downscale kernel** and **station/LR guidance** at sampling. MODS extends this by **adding more modalities**.
* **Guided inverse-problem sampling.** The sampling formulation mirrors standard **Diffusion Posterior Sampling (DPS)** style guidance (adding data-consistency gradients during the reverse pass). More inverse problem solver baselines as ablations missing.
* **Generative data assimilation (multi-modal).** Very related trends exist in **GenDA** (score-based generative assimilation from **multi-modal** satellite observations). Discussion and contrast missing.

**Questions:**

- Data availability realism. In an operational setting, station data may be delayed/noisy. How sensitive is MODS to lagged or sparse station availability during sampling? Is that sort of robustness taken into account?

- Modality importance. Is it possible to generate spatial maps of where each sensor helps (coastal, orographic, tropics)?

- Temporal collocation & missingness. How did you handle asynchronous or missing satellite scans? Any learned imputation in the encoders? What fraction of training samples include all modalities, and how does performance degrade when some are missing?

---

### Meta-Review · Area_Chair_MV1x · 2025-12-21

**Summary:**

The paper introduces a downscaling approach for meteorological data. The reviewers find the problem and method interesting, but raise concerns including similarity to previous work. No author rebuttals are provided. Therefore, I must recommend rejection.

**Reviewer Concerns:**

N/A (no rebuttals)

**Reviewer Scores:**

No changes expected, since no author responses.

---

### Decision · Program_Chairs · 2026-01-26

Reject